# Noninvasive continuous monitoring versus intermittent oscillometric measurements for the detection of hypotension during digestive endoscopy

**Anh-Dao Phan**[1,2]*, **Arthur Neuschwander**[1,2], **Guillaume Perrod**[2,3], **Gabriel Rahmi**[2,3], **Christophe Cellier**[2,3], **Bernard Cholley**[1,2]*

**1** Department of Anesthesiology and Critical Care Medicine, Hôpital Européen Georges Pompidou, Assistance Publique-Hôpitaux de Paris, Paris, France, **2** Université Paris-Descartes, Sorbonne Paris Cité, Paris, France, **3** Department of Digestive Endoscopy, Hôpital Européen Georges Pompidou, Assistance Publique-Hôpitaux de Paris, Paris, France

* phanhdao@gmail.com (ADP); bernard.cholley@aphp.fr (BC)

## Abstract

### Background

Hemodynamic monitoring during digestive endoscopy is usually minimal and involves intermittent brachial pressure measurements. New continuous noninvasive devices to acquire instantaneous arterial blood pressure may be more sensitive to detect procedural hypotension.

### Purpose

To compare the ability of noninvasive continuous monitoring with that of intermittent oscillometric measurements to detect hypotension during digestive endoscopy.

### Methods

In this observational prospective study, patients scheduled for gastrointestinal endoscopy and colonoscopy under sedation were monitored using intermittent pressure measurements and a noninvasive continuous technique (ClearSight™, Edwards). Stroke volume was estimated from the arterial pressure waveform. Mean arterial pressure and stroke volume values were recorded at T1 (prior to anesthetic induction), T2 (after anesthetic induction), T3 (gastric insufflation), T4 (end of gastroscopy), T5 (colonic insufflation). Hypotension was defined as mean arterial pressure < 65 mmHg.

### Results

Twenty patients (53±17 years) were included. Six patients (30%) had a hypotension detected using intermittent pressure measurements *versus* twelve patients (60%) using noninvasive continuous monitoring ($p$ = 0.06). Mean arterial pressure decreased during the procedure with respect to T1 ($p$ < 0.05), but the continuous method provided an earlier

**Data Availability Statement:** All relevant data are within the manuscript.

**Funding:** Edwards Laboratories provided the ClearSight™ sensors free of charge but had no role in designing the study, collecting or analyzing the data, writing the manuscript or participating in the decision to submit it for publication.

**Competing interests:** We have read the journal's policy and the authors of this manuscript have the following competing interests: BC has participated to advisory boards organized by Edwards. This does not alter our adherence to PLOS ONE policies on sharing data and materials.

warning than the intermittent method (T3 *vs* T4). Nine patients (45%) had at least a 25% reduction in stroke volume, with respect to baseline.

## Conclusion

Noninvasive continuous monitoring was more sensitive than intermittent measurements to detect hypotension. Estimation of stroke volume revealed profound reductions in systemic flow. Noninvasive continuous monitoring in high-risk patients undergoing digestive endoscopy under sedation could help in detecting hypoperfusion earlier than the usual intermittent blood pressure measurements.

## Introduction

Photoplethysmographic techniques combined with volume clamp can provide noninvasive continuous pressure (NICP) monitoring in patients. Although the validation studies have reported conflicting results regarding the interchangeability of these techniques with invasive ones [1–6], several authors consider that discrepancies in measurements should not prevent the physicians to investigate the potential advantages of continuous as opposed to intermittent pressure measurements [7, 8].

Continuous blood pressure monitoring involves arterial catheterization, which is either considered too invasive or simply too cumbersome to be used in some patient populations. Patients undergoing digestive endoscopy for example, are almost never offered an invasive pressure monitoring because the procedure is minimally invasive and its duration is usually short. The number of subjects undergoing combined upper gastrointestinal (GI) endoscopy and colonoscopy is important, since it is one the most frequent interventions performed under anesthesia in Western countries [9]. The current hemodynamic monitoring recommended by the European Society of Anesthesiology for endoscopic procedures includes continuous electrocardiography, pulse oximetry and automated noninvasive intermittent pressure (NIIP) measurements using the oscillometric technique [10]. Arterial pressure values are usually obtained at discrete intervals of five minutes. Hypotension, frequently observed during the procedure, is usually attributed to the vasodilating effect of hypnotic drugs [11–13]. However, gas insufflation used for luminal distension to facilitate gastric and colonic exploration, may potentially impair venous return and impact systemic hemodynamics.

Whether NICP increases the likelihood to detect threatening drops in arterial blood pressure in comparison to NIIP monitoring is not well established. Our primary endpoint was to compare the time-course of mean arterial pressure as obtained using the ClearSight™, a NICP device, with values obtained using NIIP from the anesthesia record sheet of patients undergoing combined upper GI endoscopy and colonoscopy. Our secondary endpoint was to analyze the variations in stroke volume estimated by the ClearSight™ using a pulse contour algorithm, during these procedures.

## Materials and methods

### Patients

This observational study was performed at a French University hospital. The institutional review board of the French Society of Anesthesiology and Intensive Care Medicine (Comité d'Ethique de la Recherche en Anesthésie-Réanimation) approved the study (IRB 00010254–

2016–014). Written informed consent was obtained from all individual participants included in the study before the procedure. Non-consecutive patients undergoing combined upper GI endoscopy and colonoscopy under sedation were prospectively included when investigators and study devices were available. Exclusion criteria were: age under 18, pregnancy, emergency procedure, sepsis, hemodynamic failure, preexisting severe cardiac comorbidity, need for mechanical ventilation, and use of carbon dioxide insufflation for the procedure. This was an exploratory study, and therefore the estimation of the number of subjects to treat was not applicable.

### Anesthetic management

Upon arrival in the procedure room, all patients were monitored using a 3-lead electrocardioscope, pulse oximetry and noninvasive intermittent arterial pressure acquired every 5 minutes. In addition, noninvasive continuous arterial pressure measurement was obtained simultaneously using the ClearSight™ monitor (Edwards Lifesciences). The arm-cuff and the digital sensor were placed on the right arm which was the upper arm when the patient was placed in left lateral decubitus for gastric endoscopy. Anesthesia was carried out with a bolus of intravenous propofol (1.5 mg/kg) followed by a continuous infusion (10 mg/kg/hr). If deemed necessary, the anesthesiologist in charge could reduce the infusion rate or titrate additional propofol (0.5 mg/kg boluses) to keep the patient comfortable, while maintaining spontaneous ventilation. No other drug was prescribed for pre-medication or co-sedation. The anesthesiologist was not aware of the noninvasive continuous pressure information, which was collected by an independent investigator. Intraoperative fluid resuscitation was left at the discretion of the anesthesiologist in charge.

### Combined gastrointestinal endoscopy and colonoscopy

The patient was positioned in left lateral decubitus for the upper GI endoscopy and then in supine position for the colonoscopy. Expansion of the digestive tract was obtained by air insufflation, using standard settings of the EXERA III Column (Olympus™, Japan). The total volume of air insufflated depended on procedural conditions and physician practice. Complete aspiration of gastric cavity was systematically performed at the end of upper GI exploration and colonic aspiration was achieved during endoscope withdrawal. Air pressure was not monitored routinely during procedures. However, gastroenterologists were asked to provide their subjective appreciation of the amount of air insufflated (major or moderate amount).

### Data acquisition

Demographic data including age, gender, height, weight, and ASA class were collected for all patients.

Routine monitoring consisted of intermittent recording of mean, systolic, and diastolic arterial pressure (MAP, SAP, and DAP respectively, mmHg) obtained using the oscillometric technique with a brachial cuff, heart rate (HR, beats/min), and peripheral oxygen saturation (SpO2, %) every ten minutes on the patient's anesthetic record chart by the anesthetic nurse. The noninvasive continuous monitor acquired and stored beat-by-beat: HR, MAP, SAP, and DAP. In addition, stroke volume (SV, mL) and cardiac output (CO, L/min) were estimated and averaged every 20 seconds. These values were recorded continuously by the noninvasive monitor and analyzed *a posteriori* by the investigator. For protocol purposes, investigators obtained hemodynamic parameters at five predefined periods: T1, baseline prior to anesthesia; T2, after anesthetic induction; T3, during gastric insufflation; T4, after gastrointestinal endoscopy and prior to colonoscopy; T5, during colonic insufflation.

For each predefined period, noninvasive continuous MAP value was calculated as the average of 5 consecutive cycles and stroke volume was calculated as the average of 5 consecutive values.

## Primary and secondary endpoints

The primary end-point of the study was to compare the profiles of MAP over the different time-points as obtained using the noninvasive continuous monitor and the oscillometric technique. The secondary endpoint was to compare the variations of SV over the time-course of the procedure.

## Statistical analysis

Data are presented as mean (± standard deviation) or median [interquartile range 25–75] as appropriate for continuous data, and as count (%) for categorical parameters. Comparisons between the independent groups were made using the Chi-squared or Fisher's exact tests for categorical variables, and using Student t-test or the Mann-Whitney test for quantitative parameters. Repeated measures acquired with intermittent and continuous methods were compared using two-way Analysis of Variance (ANOVA). Normality of data distribution was verified using Kolmigorov-Smirnov test prior to performing ANOVA. Pairwise multiple comparisons were performed using Student-Newman-Keuls method. All reported $p$ values are two-sided, and $p$ values of less than 0.05 were considered to indicate statistical significance. All analyses were performed using SigmaStat® (Jandel Scientific).

## Results

From March 2016 to June 2016, 43 patients were prospectively screened. Sixteen patients did not meet the inclusion criteria and 7 were excluded with a final enrollment of 20 patients. The flow of participants through the study is presented in Fig 1.

Patient characteristics at baseline and indication for endoscopy are summarized in Table 1. The median duration of the procedure was 34 [15–53] minutes. Gastroenterologists reported that insufflation of a major amount of gas was required for 10 patients (50%), while the others had moderate amounts insufflated.

Each patient received 5–7 mL/kg of lactated Ringer during the procedure. A single patient required one bolus of IV ephedrine (6 mg) to treat a hypotension episode.

Six patients (30%) had hypotension (MAP < 65 mmHg) detected using standard intermittent oscillometric pressure measurements *versus* twelve patients (60%) using noninvasive continuous monitoring ($p = 0.06$). The two-way ANOVA on pressure measurements obtained at the predefined periods revealed a difference between intermittent and continuous methods ($p < 0.05$). In addition, mean arterial pressure decreased over time compared to baseline ($p < 10^{-5}$), regardless of the method used for measurement (Fig 2). However, although there was no significant interaction between the NICP and NIIP MAP profiles, the decrease in MAP measured using NICP was detected earlier (T3 *vs* T1, $p < 0.05$) than the one obtained using NIIP (T4 *vs* T1, $p < 0.05$) (Table 2). The drop in systolic arterial pressure with respect to baseline was more important when assessed with the noninvasive continuous monitoring than with intermittent monitoring (-37 ± -17 mmHg *versus* -29 ± -17 mmHg, $p = 0.03$).

There were more hypotensive episodes detected when the gastroenterologist declared using major gas insufflation (90%) than when the insufflation was moderate (30%) ($p = 0.02$) (Fig 3).

There was a marked reduction in stroke volume during gastric insufflation (56 [26–86] mL), both when compared to baseline (78 [53–103] mL; $p < 0.01$) and after anesthesia induction (64 [36–92] mL; $p < 0.01$) (Fig 4). Stroke volume was also significantly lower following

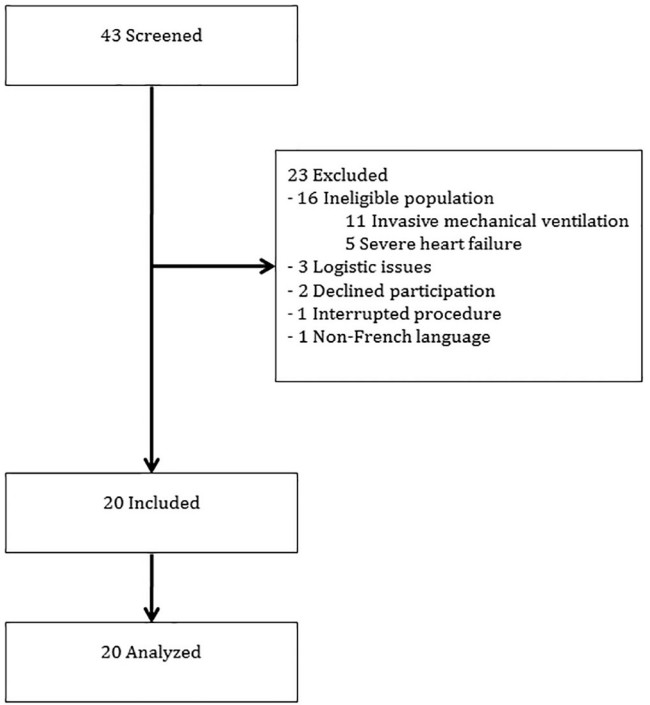

**Fig 1. Flow of participants through the study.**

**Table 1. Patients characteristics.**

| Characteristics | Patients (n = 20) |
|---|---|
| Age (yr), mean (± SD) | 53 (± 17) |
| Male sex, no. (%) | 10 (50%) |
| Body mass index (kg/m$^2$), mean (± SD) | 24 (± 4) |
| ASA physical status, mean (± SD) | 2 (± 0) |
| **Comorbidities** | |
| Arterial hypertension, no. (%) | 6 (30%) |
| Diabetes, no. (%) | 2 (10%) |
| Respiratory disease, no. (%) | 3 (15%) |
| Renal disease, no. (%) | 4 (20%) |
| Hepatic disease, no. (%) | 3 (15%) |
| Digestive surgery, no. (%) | 3 (15%) |
| **Indications for GI endoscopy** | |
| Colorectal cancer, no. (%) | 5 (25%) |
| Anemia, no. (%) | 4 (20%) |
| Abdominal pain, no. (%) | 4 (20%) |
| Gastritis, no. (%) | 3 (15%) |
| Crohn's disease, no. (%) | 2 (10%) |
| Other, no. (%) | 8 (40%) |

SD: standard deviation.

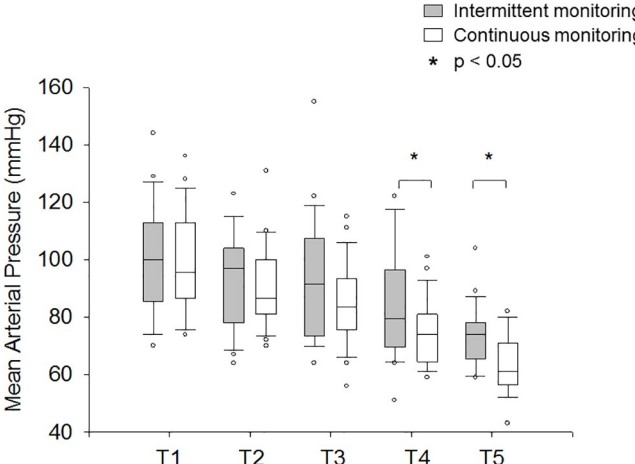

**Fig 2. Comparison between mean arterial pressure measures by standard (oscillometric) and noninvasive continuous (Clearsight™) monitoring techniques.** Data are presented as box-and-whisker plot with median, quartiles (25%-75%) and percentiles (10th-90th); circles are the data below or above those limits. T1: Prior to anesthetic induction; T2: After anesthetic induction; T3: During gastric insufflation; T4: At the end of the gastroscopy; T5: During colonic insufflation.

anesthesia induction (T2) when compared with baseline (T1) ($p < 0.01$). No significant difference was detected in stroke volume before and after colonic insufflation (Table 2). Nine of the 20 patients (45%) had a 25% reduction or more in stroke volume during the procedure, with respect to baseline.

**Table 2. Hemodynamic values obtained using noninvasive continuous (Clearsight™) and standard (oscillometric) monitoring techniques.**

| Time-points | T1 | T2 | T3 | T4 | T5 |
|---|---|---|---|---|---|
| **Noninvasive continuous monitoring** | | | | | |
| Stroke volume (mL), median [IQR] | 78 | 64 | 56 | 65 | 64 |
| | [61–86] | [55–83]* | [49–80]* † | [50–77]* ‡ | [50–70]* ‡ |
| Systolic arterial pressure (mmHg), median [IQR] | 139 | 121 | 116 | 97 | 87 |
| | [121–163] | [108–144] | [102–130]* | [82–115]* † ‡ | [75–94]* † ‡ |
| Mean arterial pressure (mmHg), median [IQR] | 95 | 86 | 83 | 74 | 61 |
| | [87–111] | [81–97] | [76–93]* | [65–81]* † ‡ | [57–69]* † ‡ |
| **Standard monitoring** | | | | | |
| Systolic arterial pressure (mmHg), median [IQR] | 136 | 131 | 115 | 102 | 100 |
| | [115–148] | [101–147] | [99–144] | [94–130]* | [87–110]* † |
| Mean arterial pressure (mmHg), median [IQR] | 100 | 97 | 91 | 79 | 74 |
| | [86–113] | [79–104] | [74–107] | [70–95]* | [66–78]* † ‡ |
| Heart rate (bpm), median [IQR] | 80 [65–92] | 80 [72–91] | 83 [75–94] | 80 [68–88] | 75 [68–80] |
| Saturation of peripheral oxygen (%), median [IQR] | 99 [98–100] | 99 [97–100] | 99 [96–100] | 98 [96–100] | 99 [97–100] |

IQR: Interquartile range. T1: Prior to anesthetic induction; T2: After anesthetic induction; T3: During gastric insufflation; T4: At the end of the gastroscopy; T5: During colonic insufflation.

* $p < 0, 01$ vs T1,

† $p < 0.01$ vs T2,

‡ $p < 0.01$ vs T3.

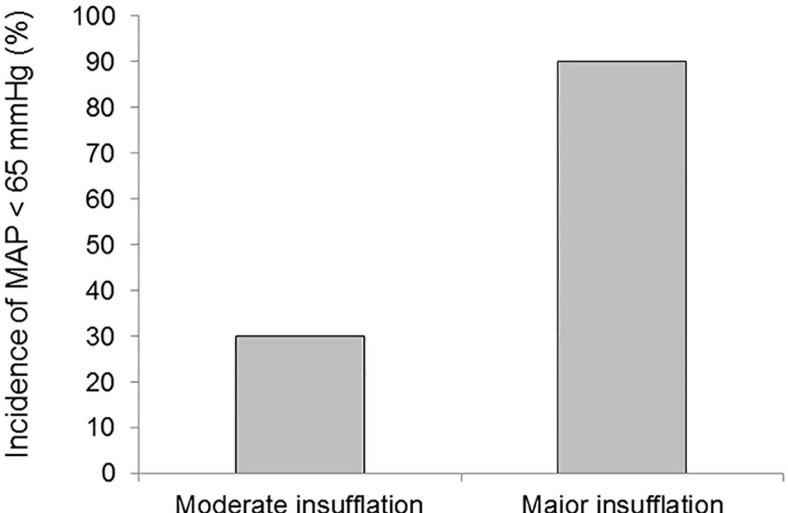

**Fig 3. Comparison of the rate of arterial hypotension measured by noninvasive continuous monitoring between moderate and major insufflation.**

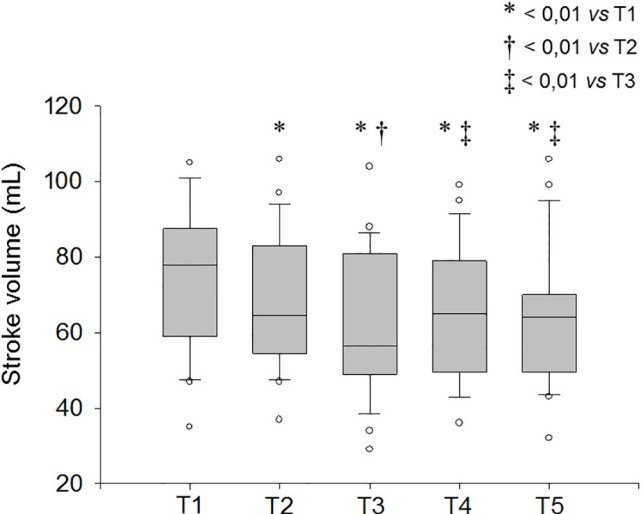

**Fig 4. Stroke volume (SV) evolution during endoscopy using noninvasive continuous monitoring.** Data are presented as box-and-whisker plot with median, quartiles (25%-75%) and percentiles (10th-90th); circles are the data below or above those limits. T1: Prior to anesthetic induction; T2: After anesthetic induction; T3: During gastric insufflation; T4: At the end of the gastroscopy; T5: During colonic insufflation.

## Discussion

In this cohort of spontaneously breathing patients, mean arterial pressure decreased significantly at the end of GI endoscopy and during colonoscopy. These variations were more pronounced according to noninvasive continuous monitoring (ClearSight™, Edwards) in comparison to intermittent oscillometric pressure measurements. Stroke volume also diminished during the procedure under propofol sedation, especially during gastric insufflation.

Hypotension during endoscopy is described as a recurrent complication reported in 5 to 32.5% of patients [14, 15]. In our cohort, 30% of the patients experienced hypotension (MAP < 65 mmHg) according to standard intermittent monitoring, but this figure increased to 60% according to noninvasive continuous monitoring. Oscillometric pressure measurement using an arm-cuff is the current recommended method to quantify arterial pressure during this procedure [10]. However, our observations confirm that the MAP variations were more frequent and more profound when the NICP was considered instead of standard NIIP. This result corroborates a previous study in patients undergoing digestive endoscopy that demonstrated a greater sensitivity of noninvasive continuous measurements for the detection of arterial pressure variations [16].

Arterial hypotension detected during upper and lower GI endoscopy appears to be multifactorial. Decreased arterial pressure is usually attributed to the vasodilating effect of sedative agents [11, 12]. Propofol is known to induce arteriolar vasodilatation [13] with a subsequent decrease in systemic vascular resistance, which in turn reduces mean arterial pressure. There is no guideline describing the "ideal" sedation technique during digestive endoscopy. We used a standard anesthesia protocol during the study to minimize the variations in arterial pressure resulting from differences in sedation regimen between patients. A bolus followed by a continuous infusion of intravenous propofol allowed for a suitable sedation during the whole procedure with minimum requirements for individual adjustments. Our observation supports the hypothesis that digestive gas insufflation impairs venous return, reduces stroke volume and could have its own role in the observed hypotension. There were, indeed, more hypotension episodes when the gas insufflation was estimated as "major" according to the operator. Stroke volume decreased after gastro-intestinal insufflation compared to anesthetic induction. This could be related to increased intra-abdominal pressure after air insufflation, which caused decreased venous return, stroke volume reduction, and further hypotension.

The clinical impact of such transient arterial pressure decrease remains uncertain. In this small exploratory study, we did not look at post-procedural morbidity. However, it is now well established that intraoperative arterial hypotension is an independent predictor of increased one-year mortality, especially in patients with pre-existing comorbidities [17–19]. We can only suspect that excess hypotension might be deleterious in patients with cardiovascular comorbidities undergoing gastro-intestinal endoscopy, but further investigation is warranted to explore this hypothesis.

This study has some limitations. First, our design is observational and descriptive, thus no causal relation between hemodynamic variations and the procedure can be inferred. Second, the ClearSight™ monitor has been compared to different current invasive devices for cardiac output measurements [8, 20–22] with acceptable reliability but this monitoring technique cannot be considered as clinically interchangeable with other measures of stroke volume [1]. Third, patient inclusion was not consecutive and was limited by investigator availability; however, our population was representative of patients undergoing endoscopy in our hospital. Finally, despite the small size of the cohort studied, we were able to detect a large difference in the detection of hypotension according to the method used for pressure monitoring.

## Conclusion

We observed that noninvasive continuous monitoring was more sensitive than intermittent oscillometric pressure measurements to detect hypotension during digestive endoscopy under sedation. Profound reductions in stroke volume could be inferred from the noninvasive arterial pressure waveforms. Similar hemodynamic alterations during surgical procedures under general anesthesia would normally prompt therapeutic interventions to improve the

determinants of tissue perfusion. Although we have no data to support an increased risk of complications in our population, the current understanding is that there might be a benefit in correcting such hemodynamic alterations. We suggest that noninvasive continuous monitoring of arterial pressure and stroke volume should be used to detect relevant hemodynamic alterations in high-risk patients undergoing digestive endoscopy under sedation. A large prospective randomized trial investigating the potential benefit of treating these alterations on the outcome of this population is desirable.

## Supporting information

**S1 File. Strobe statement.**
(DOCX)

## Author Contributions

**Conceptualization:** Anh-Dao Phan.

**Data curation:** Anh-Dao Phan.

**Formal analysis:** Anh-Dao Phan.

**Funding acquisition:** Arthur Neuschwander, Bernard Cholley.

**Investigation:** Anh-Dao Phan.

**Methodology:** Anh-Dao Phan, Arthur Neuschwander.

**Project administration:** Anh-Dao Phan, Bernard Cholley.

**Resources:** Anh-Dao Phan, Bernard Cholley.

**Software:** Anh-Dao Phan.

**Supervision:** Arthur Neuschwander, Bernard Cholley.

**Validation:** Anh-Dao Phan, Arthur Neuschwander, Guillaume Perrod, Gabriel Rahmi, Christophe Cellier, Bernard Cholley.

**Visualization:** Anh-Dao Phan, Bernard Cholley.

**Writing – original draft:** Anh-Dao Phan, Bernard Cholley.

**Writing – review & editing:** Arthur Neuschwander, Guillaume Perrod, Gabriel Rahmi, Christophe Cellier, Bernard Cholley.

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
