## [Decision Letter · Decision Letter 0]

17 Aug 2020

PONE-D-20-21784

Noninvasive continuous monitoring versus intermittent oscillometric measurements for the detection of hypotension during digestive endoscopy.

PLOS ONE

Dear Dr. Anh-Dao Phan

Thank you for submitting your manuscript to PLOS ONE. After careful consideration, we feel that it has merit but does not fully meet PLOS ONE’s publication criteria as it currently stands. Therefore, we invite you to submit a revised version of the manuscript that addresses the points raised during the review process.

I would appreciate if pay careful attention in revising your manuscript  according to the reviewer comments.  

We look forward to receiving your revised manuscript.

Kind regards,

Ehab Farag, MD FRCA FASA

Academic Editor

PLOS ONE

Journal Requirements:

2. Thank you for stating the following in the Source of funding Section of your manuscript:

"Edwards Laboratories provided the ClearSightTM sensors free of charge, but Edwards had no role in designing the study, collecting or analyzing the data, writing the manuscript or participating in the decision to submit it for publication."

"The authors received no specific funding for this work."

Additionally, because some of your funding information pertains to commercial funding, we ask you to provide an updated Competing Interests statement, declaring all sources of commercial funding.

In your Competing Interests statement, please confirm that your commercial funding does not alter your adherence to PLOS ONE Editorial policies and criteria by including the following statement: "This does not alter our adherence to PLOS ONE policies on sharing data and materials.” as detailed online in our guide for authors  http://journals.plos.org/plosone/s/competing-interests.  If this statement is not true and your adherence to PLOS policies on sharing data and materials is altered, please explain how.

Please include the updated Competing Interests Statement and Funding Statement in your cover letter. We will change the online submission form on your behalf.

"I have read the journal's policy and the authors of this manuscript have the following competing interests: Bernard Cholley has participated to advisory boards organized by Edwards Lifesciences."

Reviewers' comments:

Reviewer's Responses to Questions

**Comments to the Author**

1. Is the manuscript technically sound, and do the data support the conclusions?

Reviewer #1: Yes

2. Has the statistical analysis been performed appropriately and rigorously? 

Reviewer #1: No

3. Have the authors made all data underlying the findings in their manuscript fully available?

Reviewer #1: Yes

4. Is the manuscript presented in an intelligible fashion and written in standard English?

Reviewer #1: Yes

5. Review Comments to the Author

Reviewer #1: Interesting paper, but should be deeply corrected.

Abstract: it is confusing and not clear.

Methods.

Due to reduced sample size, check for normality should be performed.

Sample size calculation should be added

It is not clear if patients included were consecutive or not

Variations due to operators should be added

6. PLOS authors have the option to publish the peer review history of their article (what does this mean?). If published, this will include your full peer review and any attached files.

Reviewer #1: **Yes: **Fabrizio D'Ascenzo

---

## [Author Response · Author response to Decision Letter 0]

3 Sep 2020

Point-by-point responses to the editor and the reviewer

Dear Editor, dear Reviewer

We thank you for your thankful appreciation and this careful reviewing which has enabled us to greatly improve the quality of our manuscript entitled “Noninvasive continuous monitoring versus intermittent oscillometric measurements for the detection of hypotension during digestive endoscopy ”.

At your and the reviewer’s request, we in this letter diligently address every point that you have asked us to improve. Changes in the revised version of the manuscript are highlighted in red.

Responses to Editor 

1. Please ensure that your manuscript meets PLOS ONE's style requirements

Answer: We have made the necessary changes to our manuscript in order to meet PlosOne Requirements

2. Funding statement and competing interests

Answer: We have modified funding and competing statement according to the requirements. Modified funding and competing interest statements have been included in the new cover letter.

3. Competing interest section

Answer: The statement has been modified as required and indicated in the new cover letter.

4. Supporting Information

Answer: We have modified the supporting information following your instruction.

Responses to Reviewer 

1. Abstract unclear

Answer: We have revised thoroughly the abstract to clarify the message. We have also removed abbreviations as recommended. The revised abstract length is 251 words.

2. Normality check.

Answer: The normality of the data distribution (SAP, MAP, SV) was verified using Kolmogorov-Smirnov test (SigmaStat®). This precision has been added to the manuscript (page 7, lines 156-157).

3. Sample size calculation

Answer: This was an exploratory study. Prior to its completion, we had no idea of the difference of sensitivity between continuous and intermittent pressure measurements in the detection of hypotension. Therefore, we could not estimate a sample size prospectively. This was explained in the method section (page 5, line 93-95).

4. Non-consecutive patients

Answer: Patients could be enrolled in the study only when, both, investigators and equipment were available. Therefore, the patients included in this study were not consecutive. This precision has been added in the manuscript (page 4, lines 88-90)

5. Inter-operator variability

Answer: All patients were anesthetized according to our institution protocol and received similar anesthetic drugs and beseline fluid infusion. In case of hypotension, the physician in charge was free to decide to reduce the propofol infusion rate or to inject a bolus of ephedrine. Most anesthesiologists choose to modify propofol and a single patient has received a bolus of ephedrine. This is indicated page 9, line 179. 

Regarding the endoscopic management, the inter-operator variability was mainly related to the amount of gas insufflated. Since it was not possible to quantify this parameter, we choose to collect the estimation of the endoscopist regarding his subjective appreciation of the amount of gas insufflated. This was the best we could achieve in order to provide information regarding this important aspect of the procedure since (as suggested by our results) it could greatly influence systemic hemodynamics (see Figure 4).

---

## [Editor Report · Decision Letter 1]

23 Sep 2020

Noninvasive continuous monitoring versus intermittent oscillometric measurements for the detection of hypotension during digestive endoscopy.

PONE-D-20-21784R1

Dear Dr.

   Anh-Dao Phan 

We’re pleased to inform you that your manuscript has been judged scientifically suitable for publication and will be formally accepted for publication once it meets all outstanding technical requirements.

Kind regards,

Ehab Farag, MD FRCA FASA

Academic Editor

PLOS ONE
---

## [Editor Report · Acceptance letter]

25 Sep 2020

PONE-D-20-21784R1 

Noninvasive continuous monitoring versus intermittent oscillometric measurements for the detection of hypotension during digestive endoscopy 

Dear Dr. Phan:

I'm pleased to inform you that your manuscript has been deemed suitable for publication in PLOS ONE. Congratulations! Your manuscript is now with our production department. 

Kind regards, 

on behalf of

Dr. Ehab Farag 

Academic Editor

PLOS ONE